Advertisement call and genetic structure conservatism: good news for an endangered Neotropical frog

Forti Lucas R. 1 lucas_forti@yahoo.com.br
Costa William P. 2
Martins Lucas B. 3 4
Nunes-de-Almeida Carlos H. L. 1
Toledo Luís Felipe 1
1 Laboratório Multiusuário de Bioacústica (LMBio) e Laboratório de História Natural de Anfíbios Brasileiros (LaHNAB), Departamento de Biologia Animal, Instituto de Biologia, Universidade Estadual de Campinas , Campinas, São Paulo , Brazil
2 Departamento de Biologia Estrutural e Funcional, Instituto de Biologia, Universidade Estadual de Campinas , Campinas, São Paulo , Brazil
3 Faculdade de Ciências Integradas do Pontal, Laboratório de Taxonomia, Sistemática e Ecologia de Anuros Neotropicais, Universidade Federal de Uberlândia , Ituiutaba, Minas Gerais , Brazil
4 Departamento de Biologia, Programa de Pós-Graduação em Biologia Comparada, Universidade de São Paulo , Ribeirão Preto, São Paulo , Brazil
Amos William
Electronic publication date: 2016 May 10
Publication date: 2016
Volume: 4
Electronic Location ID: e2014
Received 2015 Nov 30; Accepted 2016 Apr 14
Copyright: © 2016 Forti et al.
Copyright year: 2016
Copyright holder: Forti et al.
License: This is an open access article distributed under the terms of the Creative Commons Attribution License, which permits unrestricted use, distribution, reproduction and adaptation in any medium and for any purpose provided that it is properly attributed. For attribution, the original author(s), title, publication source (PeerJ) and either DOI or URL of the article must be cited.
License URL: https://creativecommons.org/licenses/by/4.0/

Keywords: Amphibia, Anura, Bioacoustics, Call evolution, Genetic distance, Geographic distance

Funding: Fundação de Amparo è Pesquisa do Estado de São Paulo (FAPESP) 2013/09964-2, 2013/21519-4, 2013/02219-0 and 2014/23388-7 Conselho Nacional de Desenvolvimento Científico e Tecnológico (CNPq) 302589/2013-9, 152548/2011-4 and 405285/2013-2 Funding was provided by Fundação de Amparo è Pesquisa do Estado de São Paulo (FAPESP) (fellowships (2013/09964-2, 2013/21519-4, 2013/02219-0 and 2014/23388-7)); and Conselho Nacional de Desenvolvimento Científico e Tecnológico (CNPq) (fellowships (302589/2013-9, 152548/2011-4) and a research grant (405285/2013-2) to LFT). The funders had no role in study design, data collection and analysis, decision to publish, or preparation of the manuscript.

==============================
Background: Many amphibian species are negatively affected by habitat change due to anthropogenic activities. Populations distributed over modified landscapes may be subject to local extinction or may be relegated to the remaining—likely isolated and possibly degraded—patches of available habitat. Isolation without gene flow could lead to variability in phenotypic traits owing to differences in local selective pressures such as environmental structure, microclimate, or site-specific species assemblages.

Methods: Here, we tested the microevolution hypothesis by evaluating the acoustic parameters of 349 advertisement calls from 15 males from six populations of the endangered amphibian species Proceratophrys moratoi. In addition, we analyzed the genetic distances among populations and the genetic diversity with a haplotype network analysis. We performed cluster analysis on acoustic data based on the Bray-Curtis index of similarity, using the UPGMA method. We correlated acoustic dissimilarities (calculated by Euclidean distance) with geographical and genetic distances among populations.

Results: Spectral traits of the advertisement call of P. moratoi presented lower coefficients of variation than did temporal traits, both within and among males. Cluster analyses placed individuals without congruence in population or geographical distance, but recovered the species topology in relation to sister species. The genetic distance among populations was low; it did not exceed 0.4% for the most distant populations, and was not correlated with acoustic distance.

Discussion: Both acoustic features and genetic sequences are highly conserved, suggesting that populations could be connected by recent migrations, and that they are subject to stabilizing selective forces. Although further studies are required, these findings add to a growing body of literature suggesting that this species would be a good candidate for a reintroduction program without negative effects on communication or genetic impact.

Introduction

The greatest threat to endangered amphibians is habitat change caused by anthropogenic activities, which alters resource availability, environmental quality, and ecological processes (Metzger, 2001; Stuart et al., 2004). Such negative impacts have important implications for organisms that face new selective pressures exerted by habitat conversion (Forman, 1995). In addition, habitat fragmentation causes isolation of populations, and places them at risk of extinction towing to demographic stochasticity, genetic depression, social dysfunction, and exogenous factors such as strong climatic variations and disasters (Simberloff, 1986). Therefore, it is predicted that species affected by these changes would (1) migrate to appropriate adjacent areas; (2) undergo local decline and extinction; or (3) undergo local adaptation.

Surprisingly, several species thrive in modified sites even after profound anthropogenic transformation. Because these landscapes often exhibit physical, climatic, and biological (e.g., species assemblage) shifts, the ability of the remaining species to persist is likely a consequence of phenotypic plasticity in traits such as behavior, morphology, and reproduction (Mayr, 1963; Pulido & Berthold, 2004; Merckx & Dyck, 2006).

One of the most important phenotypic traits in evolutionary studies of anurans is the male advertisement call, because components of these calls are fundamental to species recognition and mate choice, and are thus under sexual selection (Ryan, 1991; Wycherley, Doran & Beebee, 2002; Smith, Osborne & Hunter, 2003; Kaefer & Lima, 2012; Grenat, Valetti & Martino, 2013). Moreover, calls are subject to natural selection over larger geographic ranges, mainly when populations are isolated by physical barriers (Simões et al., 2008; Kaefer, Tsuji-Nishikido & Lima, 2012; Tsuji-Nishikido et al., 2012). Because anurans tend to not disperse over long distances (Blaustein, Wake & Sousa, 1994; Tozetti & Toledo, 2005; Lougheed et al., 2006), sexual phenotypic traits in anurans are likely influenced by local environmental conditions (Bosch & De la Riva, 2004; Ey & Fischer, 2009).

The anuran advertisement call is a multidimensional signal that can be viewed as a collection of spectral and temporal acoustic traits that are influenced, for example, by body size, air temperature, and social context (Wells & Taigen, 1986; Gerhardt, 1991; Bee, 2002; Gerhardt & Huber, 2002; Wong et al., 2004; Toledo et al., 2015a). Therefore, sexual selection, habitat structure, and climatic conditions might all cause variation in call traits among populations (Jang et al., 2011; Faria et al., 2009; Kaefer & Lima, 2012; Kaefer, Tsuji-Nishikido & Lima, 2012; Narins & Meenderink, 2014). In addition, other biotic processes, such as interspecific acoustic interactions, which generate distinctive background noise, can affect call variation among populations (Littlejohn, 1976; Höbel & Gerhardt, 2003).

Different traits of advertisement calls may have distinct roles in anuran communication and, therefore, may evolve by distinct selective pressures (Cocroft & Ryan, 1995; Erdtmann & Amézquita, 2009; Goicoechea, De La Riva & Padial, 2010). As a result, call traits should vary in unique and predictable ways. Gerhardt (1991) classified these traits as static or dynamic acoustic traits. Typically, spectral acoustic traits show low variability (static) and are related to conspecific recognition. Consequently, static traits are subject to stabilizing or weakly directional selection by female choice. On the other hand, most temporal acoustic traits show higher variability (dynamic) and are thought to indicate a male’s investment in reproduction; these may be subject to directional selection by females for values above species means (Gerhardt, 1991; Gerhardt & Bee, 2007).

However, acoustic variation is not always related to genetic variation at the population level (Heyer & Reid, 2003; Lougheed et al., 2006; Kaefer et al., 2013). In these cases, despite some phylogenetic signal being recorded in vocalizations (Erdtmann & Amézquita, 2009; Goicoechea, De La Riva & Padial, 2010; Tobias, Evans & Kelley, 2011; Gingras et al., 2013), the evolution of genotypes and phenotypes (as acoustic traits) may be decoupled (Lougheed et al., 2006) or asynchronous (Kaefer et al., 2013).

Herein, we speculated that historical modification of landscapes by agricultural crops has created barriers among persistent populations, and affected phenotypic and genetic traits in an endangered (Assembleia Legislativa do Estado de São Paulo, 2014) Neotropical frog, Proceratophrys moratoi. We tested the hypothesis that unique selective pressures among these remaining populations have led to divergence in acoustic traits and increased genetic structure.

Methods

Species

The genus Proceratophrys includes 40 South American frog species (Frost, 2015). Proceratophrys moratoi, originally described in the genus Odontophrynus (Jim & Caramaschi, 1980), is a member of the P. cristiceps species group (Giaretta, Bernarde & Kokubum, 2000), lacking palpebral appendages and postocular swellings. It was described from the municipality of Botucatu, state of São Paulo (Jim & Caramaschi, 1980), from where it is now extirpated (Brasileiro, Martins & Jim, 2008). Despite recent reports of new populations outside Botucatu (Brasileiro, Martins & Jim, 2008; Rolim et al., 2010; Maffei, Ubaid & Jim, 2011; Martins & Giaretta, 2012), according to the current Brazilian red list, the species is endangered (EN) (Ministério do Meio Ambiente, 2014).

Proceratophrys moratoi is endemic to the Brazilian Cerrado and is found in open grasslands near small streams or swamps (Rolim et al., 2010; Maffei, Ubaid & Jim, 2011; Martins & Giaretta, 2012). Males call during the rainy season (generally from October to February). The advertisement call of P. moratoi is characterized by a single train of regularly repeated pulses. Call duration is approximately 250 ms, and the frequency ranges from 700–1,900 Hz (Brasileiro, Martins & Jim, 2008; Martins & Giaretta, 2012).

Sites

We studied six populations of P. moratoi, which represent almost its entire known geographic distribution (Martins & Giaretta, 2012). We sampled the populations in two southeastern Brazilian states (Fig. 1): São Paulo (Avaré, Bauru, Itirapina and São Carlos) and Minas Gerais (Ituiutaba and Uberlândia). These regions represent an important center of agricultural and livestock production (Ministério do Meio Ambiente, 2005) with a remarkable history of land use modification and natural habitat devastation (Dean, 1995). With the exception of the populations from Itirapina, Bauru, and Uberlândia, which were in protected natural reserves, all populations inhabited modified landscapes.

Figure 1 Geographic distribution of Proceratophrys moratoi.

State of São Paulo: 1) Avaré, 2) Bauru, 3) Botucatu (type locality–black dot), 4) Itirapina, 5) São Carlos; state of Minas Gerais: 6) Ituiutaba, and 7) Uberlândia. Map data: Google Earth, CNES Astrium, Digital Globe.

Acoustic analyses

We analyzed 349 calls from 15 males of P. moratoi from 6 localities, 18 calls from 2 males of Odontophrynus americanus and 6 calls from one male of Proceratophrys boiei. Calls were recorded using the following combinations of microphones and recorders: (1) an Audiotechnica AT 835b microphone and a Marantz PMD-222 recorder, (2) a Dynamic microphone and an Uher 4000 recorder, (3) a Sennheiser ME67/K6 microphone and a Boss 864 recorder, (4) a Sennheiser ME67/K6 microphone and a Marantz PMD671 recorder, or (5) a Sennheiser ME66/K6 microphone and an M–audio Microtrack II recorder. All recordings were made with sample rate of 44.1 or 48 kHz, and at 16-bit resolution. Acoustic recordings used in the present study are available in the Fonoteca Neotropical Jacques Vielliard, with collection numbers FNJV 10498, 10577, 12222–12224, 12228, the Smithsonian Institution website (http://vertebrates.si.edu/herps/frogs_boraceia/list.htm), and the personal collection of Ariovaldo A. Giaretta, which are detailed in the appendix of Martins & Giaretta (2012).

Acoustic analyses were performed in Raven Pro 64 1.4 for Windows (Cornell Lab of Ornithology), with the following settings: FFT (Fast Fourier Transformation) = 1,024; Overlap = 50 for spectral evaluations; and FFT = 256 and Overlap = 50 for temporal variables. Both temporal and spectral values were extracted from the spectrogram. We analyzed the following quantitative traits: frequency range, maximum frequency, minimum frequency, peak of dominant frequency, call duration, number of pulses per note, and pulse rate (pulses per second). Spectral measurements were obtained by selecting four variables in the source “choose measurements” in Raven: (1) Frequency 5% (Hz); (2) Frequency 95% (Hz)—these two measures include maximum frequency and minimum frequency, ignoring 5% below and above the total energy in the selected call; (3) Bandwidth 90% (Hz)—frequency range that included 90% of the energy distribution, i.e., the difference between Frequency 95% and Frequency 5%; (4) Max Frequency (Hz)—peak of dominant frequency (the frequency in which the power is maximum within the call). For temporal properties, we made precise selections on calls in the spectrogram, and visually counted the pulses.

We calculated the variation in quantitative acoustic variables through the coefficient of variation (CV; SD/mean) for both the among-males and within-males level. As defined by Gerhardt (1991), CVs can be used to determine if a call trait is static (CV < 5%) or dynamic (CV > 12%).

Genetic analyses

Liver and muscle samples from 26 P. moratoi individuals from six populations were collected: four populations in the state of São Paulo: Itirapina (n = 5), São Carlos (n = 5), Bauru (n = 5), and Avaré (n = 2); and two populations in the state of Minas Gerais: Ituiutaba (n = 4) and Uberlândia (n = 5). This small sample size is in part attributable to the rarity of the species, and collection restrictions, as it is endangered and apparently extinct in at least two populations. Tissues were preserved in 95% ethanol. Voucher specimens were deposited in the Coleção Científica Jorge Jim indexed in Museu Nacional, Rio de Janeiro, Brazil (CCJJ 7925, 7928–7938, 7944, 7950–7952, 7958). Tissue collection can be found in the Collection of tissue and chromosome preparation Shirlei Maria Recco Pimentel, Universidade Estadual de Campinas (UNICAMP), Campinas, São Paulo, Brazil (SMRP 469.01–469.14, 469.26–469.42). Total genomic DNA was extracted according to Veiga-Menocello et al. (2014). We targeted a 650-bp region of the 16S mitochondrial gene using the primers 16Sar and 16Sbr (Palumbi et al., 1991). Fragments were purified using a purification kit (GE Healthcare Life Science, São Paulo, SP, Brazil); sequences were obtained using the same primers and BigDye™ 3.1 cycle sequencing kits (Applied Biosystems Foster City, CA, USA), and were read on an ABI 3700/Prism. Sequences were checked by eye using BioEdit v.5.0.9 and aligned with Muscle (Edgar, 2004). Genetic distances (p distances) were computed from mitochondrial loci using MEGA 5.1 (Tamura et al., 2011).

We verified haplotypes using DnaSP v. 5.10.01 (Librado & Rozas, 2009). We obtained a haplotype network using the Median-joining network method (Bandelt, Forster & Rohl, 1999) with NETWORK 4.6.1.2.

Statistical analyses

We performed cluster analyses based on acoustic similarities using the Bray-Curtis index, through the UPGMA method and bootstrap with 1,000 randomizations (see Toledo et al., 2015b). We calculated the values for Euclidean distance among populations for acoustic traits, and correlated them with the values for geographical and genetic distance through Mantel tests with 1,000 permutations. Geographical distance was estimated in Google Earth as the straight-line distance between two sites. We conducted statistical analyses in Past 2.17 (Hammer, Harper & Ryan, 2001).

Results

Acoustic similarities and variability

The structure of calls from all individuals presented the same pattern of a single periodic pulse train (Fig. 2A). However, we found slight differences in spectral and temporal traits among calls from distinct localities (Table 1). Male calls from Avaré had the lowest frequencies, whereas calls from Ituiutaba had the highest frequencies. The individual from São Carlos presented the longest calls, whereas the individual from Bauru emitted the shortest calls. Males from Itirapina emitted calls with the highest pulse rates; the call from the male from Avaré had the lowest pulse rate. Temporal traits of the advertisement call presented high coefficients of variation among males (above 12%), and were considered dynamic (Fig. 2B). Among the spectral traits, frequency range showed the highest coefficient of variation among males, whereas the other three spectral traits presented an intermediate variation (between 5 and 12%; Fig. 2B). All call traits presented low variation within males, with CVs of lower than 8% (Fig. 2C), and the majority was considered static, with < 5% variation.

Figure 2 Call traits of the frog Proceratophrys moratoi.

(A) Waveform of the call. The call is composed of a single pulse-train structure; (B) among-male and (C) within-male coefficients of variation of advertisement traits. The horizontal continuous line represents the lower limit for dynamic traits (above 12%) in (B) and the dashed line represents the limit for static acoustic traits (below 5% of variation) in (C). Dynamic and static traits according to Gerhardt (1991).

Table 1 Acoustic traits (mean ± SD, range) of seven populations of Proceratophrys moratoi from southeastern Brazil and two close species as outgroups.

Data from the population of Botucatu were extracted from Brasileiro, Martins & Jim (2008).

Groups	Population (n = calls, M = males)	Frequency range (Hz)	Minimum frequency (Hz)	Peak of dominant frequency (Hz)	Maximum frequency (Hz)	Call duration (s)	Pulses per note	Pulses rate (p/s)	
Proceratophrys moratoi	Avaré (n = 8; M = 1)	291 ± 20 (258–301)	980 ± 20 (947–990)	1,184 ± 23 (1,163–1,206)	1,270 ± 23 (1,249–1,292)	0.297 ± 0.01 (0.277–0.315)	20 ± 0.9 (19–22)	69 ± 1.6 (66–71)	
Bauru (n = 29; M = 1)	423 ± 23 (387–474)	1,029 ± 13 (990–1,034)	1,314 ± 73 (1,077–1,378)	1,452 ± 20 (1,421–1,464)	0.227 ± 0.02 (0.160–0.260)	21 ± 2.2 (15–24)	92 ± 2.8 (81–96)	
Botucatu (n = 59; M = 2)	730	928	1,348 ± 86.6 (1,153–1,420)	1,659	0.207 ± 17.6 (146–238)	17.5 ± 1.5 (12–20)	–	
Itirapina (n = 78; M = 3)	353 ± 32 (281–388)	1,092 ± 34 (1,077–1,206)	1,317 ± 38 (1,265–1,421)	1,445 ± 27 (1,406–1,507)	0.245 ± 0.02 (0.183–0.288)	23 ± 2.4 (17–27)	94 ± 3.2 (85–103)	
Ituiutaba (n = 54; M = 2)	433 ± 66 (301–517)	1,129 ± 57 (1,077–1,249)	1,440 ± 26 (1,378–1,464)	1,562 ± 19 (1,550–1,593)	0.240 ± 0.01 (0.196–0.263)	19 ± 2 (14–22)	81 ± 4.9 (67–87)	
São Carlos (n = 26; M = 1)	288 ± 20 (258–301)	1,206 ± 0 (1,206)	1,386 ± 24 (1,335–1,464)	1,494 ± 20 (1,464–1,507)	0.307 ± 0.02 (0.274–0.382)	25 ± 1.4 (23–28)	83 ± 4.5 (71–89)	
Uberlândia (n = 141; M = 7)	343 ± 95 (215–474)	1,054 ± 70 (947–1,206)	1,286 ± 90 (1,120–1,464)	1,397 ± 92 (1,249–1,550)	0.262 ± 0.03 (0.186–0.316)	18 ± 1.3 (15–22)	71 ± 9.5 (60–97)	
Outgroup	Proceratophrys boiei (n = 5; M = 1)	577 ± 38 (517–603)	474 ± 0 (474)	637 ± 19 (603–646)	1,051 ± 38 (990–1,077)	0.743 ± 0.05 (0.666–0.795)	32 ± 1.4 (30–34)	43.1 ± 1.1 (42–45)	
Odontophrynus americanus (n = 15; M = 2)	287 ± 24 (234–328)	681 ± 24 (656–703)	825 ± 43 (750–890)	968 ± 38 (937–1,031)	0.664 ± 0.10 (0.508–0.816)	57 ± 4.6 (49–65)	86 ± 9.5 (79–103)	

Cluster analysis placed O. americanus and P. boiei as outgroups in relation to the focal P. moratoi individuals (Fig. 3). Individuals were not grouped by population, except for the males from Itirapina. The male from Avaré had the most distinct call, and was grouped with two individuals from Uberlândia. However, other males from Uberlândia were placed in the other two major groups, and individuals were not organized according to geographical distance among populations. This was confirmed by the absence of a correlation between geographical distance and the acoustic distance between populations (r = −0.23; p = 0.73).

Figure 3 Dendrogram of two outgroup species (other Odontophrynidae) and 15 males of Proceratophrys moratoi from different localities resulting from a hierarchical cluster analysis based on similarity in call traits.

Haplotype network and genetic distance

We found 7 haplotypes in the 26 partial sequences of the mitochondrial 16S gene (Fig. 4). Most haplotypes (H1–H4) were shared among multiple populations, but three haplotypes (H5–H7) were found in one individual each, and were limited to Uberlândia (H5, H6) or Ituiutaba (H7). Genetic distances of P. moratoi averaged 0.2% (0.0–0.4) between populations (Table 2) and 0.25% (0.0–0.5) within populations. Acoustic and geographical distance between populations was not correlated with genetic distance (r = −0.32; p = 0.86; and r = −0.32; p = 0.87).

Figure 4 Haplotype network of Proceratophrys moratoi populations.

The size and color of each ellipse indicate the frequency and geographic origin of the individuals.

Table 2 Genetic distances (p-distance) based on 16S mitochondrial genes between individuals of six Proceratophrys moratoi populations in the upper matrix and the respective geographic distance (in km) in the lower matrix.

Interpopulation variation (%)	Intrapopulation variation (%)	
	Itirapina	São Carlos	Bauru	Avaré	Uberlândia	Ituiutaba	
Itirapina	–	0.2	0.4	0.2	0.2	0.0	Itirapina	0.0	
São Carlos	21.65	–	0.3	0.3	0.2	0.3	São Carlos	0.1	
Bauru	114.54	116.59	–	0.2	0.4	0.4	Bauru	0.2	
Avaré	129.93	141.36	59	–	0.4	0.3	Avaré	0.5	
Uberlândia	354.97	331.88	375.50	437.27	–	0.3	Uberlândia	0.3	
Ituiutaba	392.80	372.92	375.45	437.25	119.44	–	Ituiutaba	0.1	

Discussion

In the present study, we found a common structural pattern (a single periodic pulse train; Fig. 1) for all individuals, which is consistent with the findings of previous reports on the advertisement call of P. moratoi (Brasileiro, Martins & Jim, 2008; Martins & Giaretta, 2012).

The variability in the advertisement call of P. moratoi follows a general pattern among anurans; spectral traits exhibit lower variation than do temporal ones, with the exception of frequency range (Gerhardt, 1991; Gerhardt & Huber, 2002). The low coefficients of variation in call traits we observed among males (showing highly stereotyped signals) could be attributed to stabilizing selection (Kaefer & Lima, 2012; Kaefer, Tsuji-Nishikido & Lima, 2012), which is usually a result of generalized female choice (Jennions & Petrie, 1997). Alternatively, it could reflect the absence of selection and the presence of neutral/stochastic processes (Erdtmann & Amézquita, 2009; Kaefer et al., 2013; Toledo et al., 2015b). Because sexual signals carry important information for mate recognition (Ryan, 1991), spectral traits would not be expected to diverge rapidly. Temporal traits, which are generally dynamic, have been shown to vary with social or environmental conditions (Bosch & de la Riva, 2004; Ey & Fischer, 2009). For example, some species respond to vocal interactions and chorus composition with rapid temporal adjustments in their calling behavior (Schwartz, 2001). Therefore, temporal traits such as pulse rate and call duration may be affected by social context. In species wherein females make choices based on temporal traits alone (Littlejohn, 1965), these temporal traits would be predicted to minimize the patterns observed in spectral traits. However, we observed that mate choice by acoustic properties remains to be tested, since we do not know how female P. moratoi individuals select males.

Habitat structure, background noise, and other environmental differences are pivotal in the evolution of acoustic communication in frogs (Goutte, Dubois & Legendre, 2013; Schwartz & Bee, 2013). These factors certainly contribute to regional divergence in call traits among individuals (Amézquita et al., 2006). Each reproductive environment could present distinct species composition and considerably different acoustic qualities. Based on this ecophenotypic hypothesis, we expected that local pressures would modulate call features in P. moratoi males from distinct localities, because populations of this threatened species surrounded by human-transformed landscapes could show low connectivity. In contrast, cluster analyses failed to group individuals by population (geographical context). Taken together, these results indicate minimal pressure for signal divergence, which could be explained by the following, non-exclusive hypotheses: (1) the reproductive environment in the sampled localities is similar with respect to habitat structure and background noise; (2) the female choice drives stable selection, which equalizes the general acoustic features of males from different localities; (3) the populations were recently connected, presenting traces of recent genetic flow; and (4) random evolutionary processes act on the calls (Toledo et al., 2015b). We did not test these hypotheses, but it is unlikely that populations were recently connected, as genetic distance was not correlated with acoustic distance among populations. Furthermore, although some studies have reported a correlation between genetic and acoustic distances (Smith, Osborne & Hunter, 2003; Amézquita et al., 2009), many others have shown that geographical variation in sexual signals and genetic distances among populations do not co-vary (Heyer & Reid, 2003; Lougheed et al., 2006; Pröhl et al., 2007). In these cases, it is possible that evolution has been decoupled for genotypic and phenotypic features (Lougheed et al., 2006). Cluster analysis demonstrated that a phylogenetic signal in anuran advertisement call (as a phenotypic trait) might not evolve as rapidly as DNA differences appear, a finding corroborated by Kaefer et al. (2013). Consequently, our results suggest that a phylogenetic signal would be apparent only when higher taxonomic levels are compared, for example, different species, genera, or families. This finding could be explained, in part, by the conservative nature of the molecular marker used—the 16S gene—which is commonly employed to separate different species (Fouquet et al., 2007; Brusquetti et al., 2014; Yang et al., 2014; Lourenço et al., 2015).

Although our dataset is limited, we observed that acoustic and genetic variation appears to be conserved among individuals distributed across human-altered landscapes. Our preliminary results showed similar call types and genotypes (also presenting low genetic divergence) among different populations. Such reduced acoustic and possible genetic structure could be considered in future conservation actions; for example, these findings suggest that communication barriers (prezygotic) would pose no obstacle to reproduction (Dobzhansky, 1951; Tucker & Gerhardt, 2011) if a reintroduction program were initiated for this species. Nevertheless, playback experiments are required for testing this hypothesis first. From the genetic point of view, if the actual molecular marker used in the present study could represent the genomes of all individuals, the genetic barrier (postzygotic) would pose no obstacle to reproduction too (Dobzhansky, 1951; Tucker & Gerhardt, 2011). The type locality of this species (Botucatu) is still preserved, and a reintroduction could be considered after complementary genetic, natural history, and experimental research. The causes of the decline of this population are unknown, and therefore, a reintroduction initiative could also aid in understanding past decline (if the cause is still active) and help to prevent further decline here (for example, of Bokermannohyla izecksohni—another species that could be threatened in Botucatu; L. F. Toledo & C. Z. Torres, 2015, unpublished data) and elsewhere. Thus, the findings of the present study add to the growing body of literature supporting P. moratoi as a potential candidate for conservation actions, although additional work is necessary before an action plan could be initiated.

Ariovaldo A. Giaretta recorded specimens in Uberlândia and Ituiutaba. We are grateful to Luciana B. Lourenço and Shirlei M. Recco-Pimentel for their support with the genetic analysis. Pertinent comments and English language review were provided by Kristine Kaiser from the Society for the Study of Amphibians and Reptiles (SSAR).

Additional Information and Declarations

Competing Interests

Author Contributions

Animal Ethics

Data Deposition

The authors declare that they have no competing interests.

Lucas R. Forti conceived and designed the experiments, performed the experiments, analyzed the data, wrote the paper, prepared figures and/or tables.

William P. Costa performed the experiments, contributed reagents/materials/analysis tools, reviewed drafts of the paper.

Lucas B. Martins contributed reagents/materials/analysis tools, reviewed drafts of the paper.

Carlos H. L. Nunes-de-Almeida contributed reagents/materials/analysis tools, prepared figures and/or tables, reviewed drafts of the paper.

Luís Felipe Toledo conceived and designed the experiments, reviewed drafts of the paper.

The following information was supplied relating to ethical approvals (i.e., approving body and any reference numbers):

Ethics approval was not necessary, because we analysed sound, tissues and animals already deposited in collections.

The following information was supplied regarding data availability:

Acoustic recordings used in the work are available in the Fonoteca Neotropical Jacques Vielliard, with collection numbers FNJV 10498, 10577, 12222–12224, 12228, the Smithsonian Institution website (http://vertebrates.si.edu/herps/frogs_boraceia/list.htm), and the personal collection of Ariovaldo A. Giaretta, which are detailed in the appendix of Martins & Giaretta (2012).

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
