# Peer review of "Advertisement call and genetic structure conservatism: good news for an endangered Neotropical frog"

_PeerJ, doi:10.7717/peerj.2014_

## Round 0.1 · original submission · Major Revisions

Both reviewers find the data interesting and both suggest major revision. I concur on both fronts. There are two large problems. First, the sample size is very small and poorly described. Is it not possible to sample non-destructively, given that the species is described as endangered? If not, then the limitations on sample size need to be described properly, along with the experimental design, explaining carefully why the sample size is deemed adequate. Having destructively sampled an endangered species and obtained a very small sample it is imperative to make as much as possible of the samples. To me this goes beyond sequencing a short region of the mtDNA. I would really like to see nuclear markers added as well, so that some assessment can be made of the importance of maternally directed site fidelity.

The second big issue is statistical analysis. Effectively all the comments are qualitative (“x is larger than y”). All comparisons need to supported by proper statistical tests. Trends should only be described as biologically meaningful if they are statistically significant: any measurement will vary between samples so it is inevitable that some samples will come out top and others bottom. In the one case where a test is presented, it is not described properly and appears to be the wrong one (pairwise distances cannot be correlated directly due to non-independence so need to be analysed using a Mantel test, which is here a very weak test due to the small number of populations.

In short, I would like to see nuclear makers added, or a strong case made as to why they cannot be added, and the whole analysis needs to be repeated with adequate advice from a statistician. In addition, the Referees make a number of very helpful suggestions for improvement.

Reviewer 1 ·

Basic reporting

The authors investigated variation of male advertisement calls and mitochondrial haplotypes of some populations of endangered Neotropical toad, Proceratophrys moratoi. As the result, they didn't find significant diversities both in calls and haplotypes among the populations. I agree that the methods and results described in this manuscript is fair and making sense. However, their assumption and interpretation are skewed. because they assumed existence of selective pressure and interpreted their results only from the perspective of selection without taking any notice of neutral random drift. As they gave multiple examples, there are some selective forces on advertisement calls in some cases, but that depends on the species and their surrounding environments.
They also argued these results are good for the conservation action such as reintroduction program to the type locality. However, I seriously doubtful on that because the advertisement calls is only a single part of phenotypic characters of the organism. There are some more issues on reintroduction and/or relocation over the compatibility of evolutionary significant units and introduction of diseases along with individuals. Further, they argued that there are frequent migration rates among the populations because they found little nucleotide divergence of 16S rRNA haplotypes. But the mitochondrial molecular marker is haploid and mother inheritance, and 16S rRNA gene is comparatively conservative gene in mitochondrial genome and thus slower mutation rate. Therefore the molecular marker is inadequate to discuss gene flow and migration among local populations. Little diversity of haplotypes among the populations should not be result of gene flow but just an few mutation accumulation due to younger isolation time of the populations.
In my subjective view, they seem to be challenging to use big words even based on the negative results. I recommend to rewrite the manuscript amenably on the results.

Experimental design

Their advertisement call assessment is adequate, but genetic analyses based on mitochondrial 16S rRNA gene is not.

Validity of the findings

The results itself is fine. But their interpretation is somewhat skewed.

Reviewer 2 ·

Basic reporting

There are numerous grammatical errors and awkward constructions.

Experimental design

I was not convinced that the sample size for the genetic analyses was sufficient for this study. I was also not convinced that the markers used were appropriate.

Validity of the findings

Given the problems with the sampling, markers and analyses, I did not find the conclusions convincing.

Additional comments

Review: Neotropical frogs (PeerJ)

Overall, I thought this was an interesting study. However, I did have some concerns that I think should be addressed before this work is suitable for publication. Below I provide comments on specific parts of the manuscript that highlight these issues.

L57: Here and many other places there are awkward constructions and grammatical errors. I know this not always easy, but I encourage the authors to go through the manuscript with someone who can spot and correct these errors thoroughly and effectively. In this case, the phrase “by evolution they undergo local adaptation” is awkward and confusing. I would suggest simply “undergo local adaptation”.

L74: “package” is not an appropriate descriptor here – please find an alternative.

L88: “trait present higher…” another awkward sentence – rewrite.

L98: What historical modifications? You need to be specific here.

L163: The sample sizes seem extremely small for this type of population genetic analysis. If this was due to limits on collecting imposed by conservation considerations the authors should point this out specifically.

L186: UPGMA suffers from a number of drawbacks, and it is not clear why the authors chose to use this method. At the least, the authors should explain and defend the rationale for this choice. I would prefer to see the authors repeat the analysis with one or more alternative methods to see if the analysis yields similar results.

L203 and L227: I must be missing something, but if frequency range shows the highest coefficient of variation, how is this consistent with the general pattern, if the general pattern is for spectral traits typically have low acoustic variability (see Line 86). I think further explanation is in order here.

L242: “and then mate choice” – awkward construction – rewrite.

L247: “Two different reproductive…” awkward sentence – rewrite.

L248: “Based in this…” change to “Based on this…”

L251: “present” is an awkward word in this context – choose another.

L251: “In contrast…” confusing sentence – rewrite for clarity.

L256: “the stabilizing selection by female choice” – awkward construction – rewrite.

L260: change “have been recently connected” to “were recently connected”

L261: “although some studies…” Frankly, given the small sample sizes of individuals analyzed and the extremely conservative genetic markers used, I don’t think you can say anything definitive concerning the relationship between genetic variation and acoustic variation, or genetic variation in relation to “local adaptation” and the potential for reintroduction. The region of mtDNA sequenced (part of the 16S gene) is notoriously conservative, so the fact that minimal distances were found between populations is not surprising. This would also make it very hard to detect any associations between phenotypic differences (i.e. call parameters) and genetic variation. With respect to conservation, arguing that populations are genetically “replaceable” goes far beyond what you can reasonably conclude from these results. You would have to look at a variety of markers known to be highly variable across populations to have a remote case for this claim. Overall, I thought the choice of genetic markers for this study was not appropriate, given the conclusions proposed.

---

## Round 0.2 · Major Revisions

This paper is improved but a number of issues remain. These are as follows:

1. Given lethal sampling of an endangered species it is vital to make as much as possible of the samples that have been collected. To screen for just one, very low variability mitochondrial marker seems to me wrong. It is also incorrect to say that much larger samples of animals are needed for nuclear markers. A small panel of even just 5-10 microsatellite markers would tell us a lot more about inter-individual relatedness between individuals from the same and neighbouring sampling locations and about the levels of genetic variability present. Since the primary objection appears to be that more samples would be needed, and even though I only rarely think it is fair to ask for more experimental work to be conducted, in this case I think the authors should revisit their samples and generate at least some microsatellite data.

2. One Referee asks for improvement of the English. Again, as someone who could not write a paper in another language I am rather embarrassed to write this, but this point has not been addressed to any extent. There remain many, many errors, a lot of which would be found by even a basic grammar-checker. The authors need to find a native speaker to help get the paper in better shape.

3. In general, there is a tendency to use stronger language than is needed, as well as to over-interpret their findings. For example, in line 60 they state “likely a consequence of remarkable phenotypic plasticity”. The word ‘remarkable’ is redundant to the point of being wrong: there is no evidence at all about the level of phenotypic plasticity needed, nor of how this compares to levels found in other scenarios. More accurate would be to say “Phenotypic plasticity is one factor that may play a role in allowing some species to survive while others are lost”. Superlatives like “remarkable” need to be used with great caution and are usually best avoided.

4. More important is the over-interpretation of the results. The basic finding is that, within an understandably small data set, calls and genotypes vary between populations and appear uncorrelated with each other or with geography. This is interesting, but it tells us nothing about the likely success or otherwise of possible conservation strategies such as genetic rescue. Not only are the data far too few to draw robust rather than preliminary conclusions, but mitochondrial DNA similarity tells us effectively nothing about the likely success or otherwise of introducing animals from outside the current gene pool. Statements in this area need to be accurate and cautious so that less informed readers do not use them as the basis of deeply flawed conservation policies. I think the whole last paragraph goes too far and I would look to summarise using a much simpler, clearer statement like: “Genetic augmentation may at some time be considered in this species. Our limited study provides no evidence that such action would be hampered by genetic or communication barriers, though much more work would be needed before a program should be initiated”.

---

## Round 0.3 · accepted · Accept

We have received re-reviews which are generally positive and as a result I have decided to accept the paper, even though my personal opinion is that the case is extremely borderline and the paper could probably be rejected on the grounds that it combined minimal sample sizes with minimal genotyping effort. As I said before, it should be a general principle that lethally sampled animals should be analysed as fully as possible, particularly when the species is endangered. The splitting of the genetics into different bits that then 'have' to be published separately means that neither paper will be as good or robust as it should be: the desire to publish more papers has taken precedence over the quality of science. I hope the authors will bear this in mind in any similar future studies and will do their samples better justice by sequencing more of the mtDNA (multiple regions, including the D-loop) and, where necessary, negotiating harder for access to the microsatellite data. Please note, the English still needs some polishing.

Reviewer 1 ·

Basic reporting

Forti et al. evaluated genetic and acoustic diversities of endangered frog, Proceratophrys moratoi in Brazilian Cerrado. Though their materials and method used are not enough to conclude comprehensive genetic structure of the focal species, their results have enough value to be published especially when considering conservation status of the species (“CR” in IUCN Redlist).
However I still have some comments in some of their statements as written below.

The content of subsection “Species” which introduced P. moratoi should be included in Introduction being relevant to the general background of conservation and diversity of organisms.

P210: Why the samples from Avare could be the “most distinct call”? It should be mentioned more precisely.

P239-P244: By merely mentioning some cases in the other species, the authors argued that the temporal traits are under the effect of rather social context than environmental conditions. I do not believe the rationale of this statement.

Experimental design

It could not be say the method and materials used are fully sufficient. But, considering conservation status of focal species, it is unavoidable.

Validity of the findings

No Comments.

Reviewer 2 ·

Basic reporting

I think the authors have done a good job of reporting their results.

Experimental design

The experimental design is appropriate given the limitations inherent in the system. I think the authors should have used different genetic markers (and this was discussed in my previous review), but I think the data they have acquired is acceptable and worth publishing.

Validity of the findings

The findings are valid, and the authors have toned down their conclusions and discussion on the basis of previous reviewer comments.

Additional comments

Overall, I think the manuscript if substantially improved, and I would recommend publication after making a few minor revisions.

Abstract - Methods - change "genetic distance" to "genetic distances", and change "by the haplotype network" to "with a haplotype network analysis"

Abstract - Results - second sentence is awkward - rewrite for simplicity and clarity.

Abstract - Discussion - last sentence - remove the word "individuals", replace with "a"

Methods - Acoustic Analyses - last paragraph - change "through coefficient" to "through the coefficient"and change "for both among-males" to "for both the among-males"